# Functionalized Crystalline *N*-Trimethyltriindoles: Counterintuitive Influence of Peripheral Substituents on Their Semiconducting Properties

**DOI:** 10.3390/molecules27031121

**Published:** 2022-02-08

**Authors:** Sergio Gámez-Valenzuela, Angela Benito-Hernández, Marcelo Echeverri, Enrique Gutierrez-Puebla, Rocío Ponce Ortiz, Maria Carmen Ruiz Delgado, Berta Gómez-Lor

**Affiliations:** 1Department of Physical Chemistry, University of Málaga, Campus de Teatinos s/n, 29071 Málaga, Spain; sergiogamez@uma.es (S.G.-V.); rocioponce@uma.es (R.P.O.); 2Instituto de Ciencia de Materiales de Madrid-Consejo Superior de Investigaciones Científicas (ICMM-CSIC), Sor Juana Inés de la Cruz 3, Cantoblanco, 28049 Madrid, Spain; angelabenitohernandez@gmail.com (A.B.-H.); marceloeche525@hotmail.com (M.E.); egutierrez@icmm.csic.es (E.G.-P.)

**Keywords:** triindole, organic semiconductors, DFT-calculations, OFETs, Raman

## Abstract

Three crystalline *N*-trimethyltriindoles endowed with different functionalities at 3, 8 and 13 positions (either unsubstituted or with three methoxy or three acetyl groups attached) are investigated, and clear correlations between the electronic nature of the substituents and their solid-state organization, electronic properties and semiconductor behavior are established. The three compounds give rise to similar columnar hexagonal crystalline structures; however, the insertion of electron-donor methoxy groups results in slightly shorter stacking distances when compared with the unsubstituted derivative, whereas the insertion of electron-withdrawing acetyl groups lowers the crystallinity of the system. Functionalization significantly affects hole mobilities with the triacetyl derivative showing the lowest mobility within the series in agreement with the lower degree of order. However, attaching three methoxy groups also results in lower hole mobility values in the OFETs (0.022 vs. 0.0014 cm^2^ V^−1^ s^−1^) in spite of the shorter stacking distances. This counterintuitive behavior has been explained with the help of DFT calculations performed to rationalize the interplay between the intramolecular and intermolecular properties, which point to lower transfer integrals in the trimethoxy derivative due to the HOMO wave function extension over the peripheral methoxy groups. The results of this study provide useful insights into how peripheral substituents influence the fundamental charge transport parameters of chemically modified triindole platforms of fundamental importance to design new derivatives with improved semiconducting performance.

## 1. Introduction

In the last few years, the heptacyclic molecule 10,15- dihydro-5*H*-diindolo[3,2-*a*:3′,2′-*c*] carbazole (also known in the literature as triindole or triazatruxene) has been identified as a highly promising *p*-type organic semiconductor in the construction of electronic and optoelectronic devices. Owing to its excellent hole-transporter ability, numerous triindole-based materials have been successfully incorporated in OFETs [1,2,3,4] and dye-sensitized [5,6,7] or perovskite-based [8,9] solar cells exhibiting excellent performances. On the other hand, their favorable optical properties have been successfully exploited in the fabrication of efficient OLEDs [10,11,12,13,14], waveguiding materials [15] and photoconductors [16]. In addition, the triindole core, has been identified as an excellent aromatic core in the construction of high mobility *p*-type semiconducting discotic liquid crystals. By attaching a number of peripheral flexible chains to this molecule, it is possible to induce in this heptacyclic platform columnar mesomorphism giving rise to liquid crystalline materials that exhibit excellent hole mobilities values and can be easily processed from solution or melt [17,18,19]. Interestingly, mesomorphism can also be induced in this core by means of electronic factors. Particularly, making this core more electron-rich by peripheral substitution with electron donors such as methoxy groups, favors its self-assembly giving rise to highly ordered columnar mesophases [20]. This effect has been rationalized on the basis of predominant CH-π interactions driving the self-assembly, as these interactions are favored over increasing the electron density of the aromatic CH-acceptor system.

The supramolecular organization of triindole-based materials is also strongly affected by varying the substitution pattern on the nitrogens. We have found that attaching different-sized alkyl chains strongly influences how these molecules interact with their neighbors, with important consequences both in their solid-state packing, their mesomorphic behavior, and their semiconducting performance [21]. In particular, the presence of three *N*-methyl groups induces a columnar hexagonal arrangement due to the participation of cooperative CH-π interactions involving all the methyl groups in the crystal.

Taking advantage of the beneficial influence of *N*-methyl groups in the triindole core, in this manuscript we have synthesized two new *N*-trimethyltriindole derivatives functionalized with different substituents at 3, 8 and 13 positions (either three methoxy or three acetyl groups). We have investigated how the electronic nature of the attached peripheral substituents influences their electronic properties, solid-state organization, and semiconductor behavior. Curiously, although the trimethoxy derivative gives rise to a similar columnar hexagonal organization but with shorter stacking distances (in agreement with more favorable CH-π interactions), the mobility value decreases in comparison with that of parent *N*-trimethyltriindole. This unexpected behavior has been ascribed, with the help of density functional theory (DFT) calculations, to lower transfer integrals due to the HOMO wave function extension over the peripheral methoxy groups. We hope that this study will help to shed light on how peripheral substituents influence fundamental charge transport parameters of chemically modified triindole platforms of fundamental importance to design new derivatives with improved semiconducting performance.

## 2. Results and Discussion

### 2.1. Synthesis and Electronic Properties

Figure 1 shows the structure of the three triindole derivatives under study. The synthesis of compounds **1** [22] and **2** was performed by methylation of the triindole and 3,8,13-trimethoxytriindole [23], respectively, under basic conditions using tetra n-butylammonium hydrogen sulfate as a phase transfer catalyst and treatment with (CH_3_)_2_SO_4_. Compound **3** was obtained by a Friedel–Crafts reaction between *N*-trimethyl triindole and the acetyl chloride in the presence of a Lewis acid (AlCl_3_). Acylation occurs selectively at positions 3,8 and 13.

The electronic properties of the three compounds were studied by UV-vis spectroscopy and cyclic voltammetry and rationalized with the help of DFT and time-dependent DFT (TD-DFT) calculations. Figure 2a compares the electronic absorption spectra of the three compounds in CH_2_Cl_2_ solution. Functionalization of triindoles at positions 3,8, and 13 produces a bathochromic shift independently of the electronic nature of the substituents when compared with the non-peripherally substituted derivative **1**; note that the maximum absorption band appears at 317 nm in **1**, 332 nm in **2** and 339 nm in **3**. In addition, the absorption edge is also notably redshifted for derivatives substituted with electron-withdrawing acetyl groups, which results in the lowest optical gap within the series. In fact, the optical band gap estimated from the tangent to the low energy edge of the absorption band is redshifted in the following order: 369 nm (3.36 eV) in **1**, 401 nm (3.09 eV) in **2** and 466 nm (2.66 eV) in **3**.

Time-dependent DFT (TD-DFT) vertical excitation energies nicely reproduce the experimental data, predicting a maximum absorption band localized at the lowest energies and several electronic transitions around 250 nm with a π−π* nature (Appendix A). It should be noted that the redshift of the absorption maximum on going from **1** to **3** is well captured from the TD-DFT calculations, being the difference within the typical values reported in the literature [24]. The maximum absorption band results from the overlap of the S_0_→S_3_ and S_0_→S_4_ electronic transitions, which are assigned to different combinations of HOMO–LUMO, HOMO-1–LUMO, HOMO-1–LUMO+1 and HOMO–LUMO+1 one-electron excitations. It is worth noting that in these systems the S_0_→S_1_ transition is forbidden by symmetry with respect to dipole−dipole selection rules, but that the distortions from the *C*_3_ geometry in solution can contribute to the activation of these electronic transitions [25]. As evidenced by the frontier molecular orbitals shown in Figure 3, due to their *C*_3_-symmetric structures, the HOMO and LUMO energy levels of these triindole systems are doubly degenerated, being delocalized over the entire molecular π-frameworks.

Interestingly, the insertion of the methoxy groups in **2** causes the extension of the HOMO over these peripheral donor groups, producing a destabilization of the HOMO while the LUMO level is barely affected. On the other hand, the attachment of acetyl groups in **3** produces the extension of LUMO over these electron-withdrawing groups, and results in a moderate HOMO and LUMO stabilization with the latter in a bigger extension. Thus, the peripheral substitution of the triindole platform allows for better extension of π-conjugation, decreasing the HOMO–LUMO gap in the following order: **1** (4.30 eV) > **2** (4.17 eV) > **3** (3.97 eV), which is in good agreement with the trends observed by the optical gaps.

As shown in Figure 2b, cyclic voltammetry confirms that **1**–**3** compounds can be easily and reversibly oxidized. Interestingly, the first oxidation potentials are significantly influenced by the nature of the peripheral substituents. Peripheral substitution with electron-donor groups increases the oxidizability of the system, while functionalization with electron-acceptor groups produces a displacement of the oxidation potentials to more positive values as a consequence of the decrease in the electronic density of the π system. To evaluate the influence of the substitution on the energy levels, we have estimated their HOMO levels from the first oxidation potential referenced to the ferrocene/ferrocenium redox couple and considering a value of −4.8 eV for Fc with respect to zero vacuum level. The HOMO–LUMO gap was estimated by the absorption edge of the electronic spectra, allowing us to estimate the LUMO energy level by subtracting this value to that of the HOMO energy (see Table 1). Interestingly, all three compounds present HOMO energy values close to the Au work function (ΦAu = 5.1 eV), which makes them potential candidates for the study of their semiconductor properties through their incorporation into OFETs. The trisubstitution with electron-acceptor acetyl groups moderately influences the energy level of the LUMO orbitals without affecting the favorable value of the HOMO orbital, while the trisubstitution with electron-donor methoxy groups barely affects the frontier energy levels. These results are in excellent agreement with the DFT-calculated HOMO and LUMO energy levels mentioned above (see Table 1).

We now make use of Raman spectroscopy to evaluate the effect that the different electronic nature of the peripheral substituent plays on the π-conjugational properties of the triindole platform. It is interesting to highlight that Raman spectroscopy can provide relevant information about the electronic coupling between covalently connected conjugated moieties and the effective π-conjugation length in semiconductors [26,27]. Figure 4 compares the solid-state FT-Raman spectra of the three triindole-based systems. We observe that the Raman bands collected in the 1500−1650 cm^−1^ region, which are associated with C=C/C-C stretching modes, are selectively enhanced in the spectra; this is ascribed to the existence of an effective electron–phonon coupling that characterizes the π-conjugated systems [28,29]. Then, we pay attention to the double peak at ∼1605 cm^−1^ and ∼1570 cm^−1^, which arise from a CC stretching mode (i.e., mode 8a of benzene [30]) involving the external benzene rings and the innermost benzene ring, respectively. The band at 1605 cm^−1^ is very sensitive to the electronic nature of the substituent shifting towards higher (lower) frequency upon trimethoxy(triacetyl) functionalization, which is ascribed to the electron-donor (acceptor) character of the peripheral substituents. On the other hand, the band at 1570 cm^−1^ shows a frequency downshift going from **1** to **3**, which better reflects π-electronic delocalization upon functionalization, this effect being more relevant in the case of **3**. In addition, an increase in the intensity of the band at 1570 cm^−1^, together with that recorded at 1666 cm*^−^*^1^, which is ascribed to the stretching of the C=O groups (Appendix A) with respect to those appearing at ~1200–1300 cm*^−^*^1^ arising from non-conjugated CH_2_ bending vibrations, supports the more efficient π-conjugation in **3** in agreement with its lower band gap.

### 2.2. Crystal Structure

All three compounds are obtained as crystalline materials. The crystal structure of **1** has been previously reported by us [22]. On the other hand, the slow evaporation of a CH_2_Cl_2_ solution of **2** gives rise to colorless needle-shaped crystals of sufficient quality for single crystal structure determination. Unfortunately, we could not obtain single crystals of the triply acetylated derivative **3**; however, its powder difractogram reveals a number of reflexion peaks with a reciprocal spacing ratio of 1:√3:2:√7 indicative of a columnar hexagonal organization with an *a* parameter of 25.23 Å. Interestingly, the position of the peaks’ maxima matches very well with the simulated X-ray diffractogram of **2** obtained from its single crystal data (see Appendix A), pointing to a similar structure and packing pattern.

X-ray analysis of *N*-trimethyltriindole **1** indicates that in this case the triindole core is slightly twisted, with the three peripheral rings bending out of the plane of the central aromatic ring with dihedral angles of 8.06° and with the methyl groups slightly out of the molecular plane of the central ring in an *all-syn* conformation (distance of methyl group to the plane of the 0.42 Å). Conversely, functionalization with three methoxy groups significantly planarize the molecule: the dihedral angle between the peripheral rings and the central aromatic ring decreases to 1.86°, and the methyl and methoxy groups accommodate within the plane of the molecule.

Compound **2** crystallizes in the hexagonal space group P63/m, while **1** crystallizes in R-3 space group. As can be observed in Figure 5, a comparison of the crystal packing of the two compounds shows that in both cases molecules pack into the crystal, forming columns that extend along the crystallographic axis c, thus favoring one-dimensional macroscopic growth.

Along the columns the molecules orient face-to-face in an alternate arrangement, each molecular unit rotated by 60° with respect to its next neighbors in the stacks and with the central aromatic rings perfectly superimposed. Within the stacks, the *N*-trimethyltriindole units (**1**) are situated at two different distances that alternate along the column reflecting the steric hindrance by the methyl groups (centroid–centroid distance of the central aromatic rings of two adjacent molecules are 3.53 Å and 3.68 Å), while in the trimethoxy derivative **2** (with the methyl groups within the molecular plane) only one repeating distance is observed (3.49 Å).

In both cases, we can see that each methyl group is involved in CH ··· π interactions with the aromatic rings that lie above and below it. Please note that in the figure we have shown only the unique interactions of **2**, but due to the hexagonal symmetry of the crystals, all the methyl groups would be involved in such CH ··· π that act cooperatively in the stabilization of the column. Curiously, CH-π interactions between the α-methylenic group of the *N*-alkyl chains and the external rings of the triindole core have been previously found to stabilize similar alternated stacking in *N*-dodecyltriindole derivatives both in solution [31] and in the solid state [20].

We now make use of DFT calculations to better understand how peripheral functionalization impacts on the interplay between intramolecular and intermolecular interactions. First, the DFT-calculated isolated molecular geometries were evaluated and similarly twisted triindole platforms are found for the three systems (Appendix A). It is not surprising that the twisted triindole conformations are the most energetically stable since they contribute to relaxing the steric hindrance caused by the methyl group. Interestingly, similar energy differences between the relaxed (i.e., twisted triindole platform) and constrained-planar (i.e., the triindole platform is forced to maintain the planarity) geometries are found within the series (i.e., ~6 kcal/mol, see Appendix A), indicating that functionalization on the periphery exerts a slight impact on the energetics of the isolated molecular structure. However, in its solid state, **1** maintains the twisted triindole platforms in the crystal while the triindole backbones in **2** display a planar conformation; this points to the influence that molecular neighbors play in leading a planar conformation in the solid state overcoming the tendency to twist [32]. To corroborate this, we have performed DFT calculations including dispersion corrections for a supramolecular complex formed by three molecules extracted from the crystalline structure of compound **1**; the same model was also used for **2** upon insertion of the peripheral methoxy groups. As seen in Figure 6, the resultant optimized geometries reveal that the non-peripherally substituted triindole platforms **1** maintain the twisted conformations in the aggregate while in the trimethoxy derivative **2** the central molecule becomes planar with a shorter π−π distance between the molecules. Importantly, this effect has not been observed when the calculations are performed without including an explicit correction for dispersion (Appendix A), highlighting the role played by weak, noncovalent interactions in the stabilization of the planar structures upon methoxy substitution.

### 2.3. Electrical Characterization and Charge–Transport Parameters

In order to study the molecular structure charge–transport relationship of the triindole derivatives, their thin-film transistors (TFT) were fabricated in a standard bottom gate-top contact architecture (see materials and methods section for more details of device fabrication). The optimization of the device fabrication steps suggests that both substrate treatments and thermal annealing temperatures play a relevant role in the resultant mobilities of the devices. After thermal annealing at 60 °C (90 °C), the mobilities increased dramatically by three (two) orders of magnitude for the octadecyltrichlorosilane-treated substrated of **1** (**2**) (Appendix A). This fact is in total agreement with the progressive enhancement of crystallinity observed on the thermally treated devices based on triindole **2** (chosen as representative example), as determined by Grazing Incidence X-Ray Diffraction (GIXRD) and Atomic Force Microscopy (AFM) measurements (Appendix A). On the other hand, it was found that the performance of OTFT of **3** is relatively insensitive to thermal treatments (see Appendix A), in consonance with the amorphous nature of the films under different annealing conditions (Appendix A). The typical transfer and output characteristics for the fabricated devices under the optimal fabrication conditions are illustrated in Figure 7, and their p-channel transport parameters, including charge carrier mobilities (µ), threshold voltages (V_T_) and intensities ratios (I_on_/I_off_) extracted from the transfer plots of the devices and recorded in the saturation regime, are summarized in Table 2. The field effect mobilities of the studied semiconductors are modest, with the maximum p-type mobility of 2 × 10*^−^*^2^ cm^2^V*^−^*^1^s*^−^*^1^ for the unsubstituted triindole **1** in accordance with previously reported data [3]. Quite high on/off ratios of ~10^7^ and low threshold voltages of −13 V were measured for this semiconductor. Nevertheless, the introduction of peripheral methoxy groups in **2** impairs the electrical properties, reducing by an order of magnitude the p-type mobility (1 × 10*^−^*^3^ cm^2^V*^−^*^1^s*^−^*^1^) and by four orders of magnitude the on/off ratios (~6 × 10^3^) with respect to **1**. Comparable threshold voltages (−4 V) can be observed. Interestingly, the substitution with electron-withdrawing acetyl groups in **3** maintain the p-type polarity. However, the transistor parameters are dramatically reduced in comparison with triindole **1**, with a three-orders-of-magnitude lower field effect mobilities (3 × 10*^−^*^5^ cm^2^V*^−^*^1^s*^−^*^1^) and five-orders-of-magnitude lower on/off current ratios (~2 × 10^2^). This can be related with the amorphous nature observed for the thin films of this compound in the GIXRD measurements (Appendix A).

In order to gain insight into the origin of the different charge–transport properties and to establish structure–charge transport relationships for these triindole derivatives, we have analyzed the key parameters impacting the charge mobility at the molecular level. In this context, we have calculated the intramolecular reorganization energies associated with hole transfer (λ_h_), which reflect the geometric changes needed to accommodate a positive charge (the smaller λ_h_ the larger the expected charge mobility), and the hole transfer integral (t_h_), which characterizes the degree of intermolecular electronic interactions for hole transfer between nearest-neighbor molecular pairs. A modest increase in the λ_h_ values is found upon peripheral substitution (with λ_h_ values of 231 meV for **1**, 257 meV for **2** and 272 meV for **3**), this being in line with the slightly larger geometrical relaxation found upon oxidation in the peripherally substituted systems (Appendix A). On the other hand, smaller transfer integrals for hole transport (t_h_) are found for dimers extracted from the experimental single crystal data of **2** (28 meV) when compared to those found for dimers of **1** (71 meV). This can be ascribed to less efficient wave function overlaps in **2,** despite their more planar backbones and smaller stacking distances; note that the HOMO wave function extension over the peripheral methoxy groups (see Figure 3) might result in less favorable wave function overlap in the central core of the staggered (60° rotated) triindole platforms, resulting in smaller t_h_ values. To corroborate this point, we have also calculated the t_h_ values for a cofacial model dimer with a rotation angle of 60° between the disks and an intermolecular distance of 3.51 Å (Appendix A); whereas a 30% drop of the t_h_ values is found upon methoxy functionalization, the insertion of acetyl groups maintains the t_h_ values obtained for the non-peripherally substituted system since the HOMO wave function is not affected by the insertion of the electron-withdrawing groups. Thus, our data reveal smaller t_h_ values and slightly larger λ_h_ values for **2** when compared to **1** that might result in smaller charge–transport properties in **2,** which is consistent with the better device performance exhibited in OTFTs based on **1** than on **2** (hole mobilities of ca. 0.022 vs. 0.0014 cm^2^V^−1^s^−1^). On the other hand, the lowest mobility obtained in compound **3** cannot be explained based on intrinsic molecular factors, and it might be probably related to a less ordered columnar packing in agreement with the poor crystalline organization of it that was experimentally observed.

## 3. Materials and Methods

**Synthesis and characterization of 2.** To a solution of 3, 8, 13-trimethoxytriindole (84 mg, 0.19 mmol) in 20 mL of THF, KOH (324 mg, 5.78 mmol) and (CH_3_)_2_SO_4_ (146 mg, 1.15 mmol) are added at 0 °C. The mixture is refluxed under magnetic stirring for 1 h. The mixture is partitioned between H_2_O and CH_2_Cl_2_ and the organic phase was washed with H_2_O, dried over MgSO_4_ anhydrous and evaporated under reduced pressure. The pure product **2** (39 mg, 42% yield) was obtained from the residue by precipitation from a mixture of CH_2_Cl_2_/C_6_H_6_ followed by centrifugation as a yellow solid.

^1^H-RMN (300 MHz, CDCl_3_, 25 °C, ppm) δ 7.92 (s, 3H), 7.43 (d, *J =* 8.7 Hz, 3H), 7.08 (dd, *J =* 8.8, 2.4 Hz, 3H), 4.33 (s, 9H), 4.01 (s, 3H); ^13^C-RMN (75 MHz, CDCl_3_, 25 °C, ppm) δ 154.1, 140.2, 137.4, 123.6, 110.4, 110.1, 106.7, 102.5, 56.4, 36.2. APCI MS m/z 478 (M+H^+^); HRMS (APCI) calcd para C_30_H_28_N_3_O_3_: 478.2125, found: 478.2123.

**Synthesis and characterization of 3****.** To a solution of **1** (260 mg, 0.67 mmol) in 10 mL of dry CH_2_Cl_2_, AlCl_3_ (270 mg, 2 mmol) and CH_3_COCl (15.8 mg, 0.201 mmol) are added at 0 °C. The mixture is allowed to reach temperature and stirred for 4 h. H_2_O is added and the resulting solid is filtered and washed thoroughly with H_2_O, CH_3_CN and CH_2_Cl_2_ to give **3** as a yellow solid (206 mg, 60%yield).

^1^H-RMN (300 MHz, C_2_D_2_Cl_4_, 100 °C, ppm) δ 9.22 (s, 3H), 8.17 (m, 3H), 7.66 (m, 3H), 4.60 (s, 9H), 2.80 (s, 9H). ^13^C-RMN (75 MHz, C_2_D_2_Cl_4_, 100 °C, ppm) δ 197.1, 144.8, 140.0, 130.4, 124.6, 123.0, 122.41, 109.3, 104.0, 36.1, 26.4. APCI MS m/z 514 (M^+^+1); HRMS (APCI) calcd para C_33_H_28_N_3_O_3_: 514.2025, found: 514.2129

**FT-Raman spectra** were obtained with an FT-Raman accessory kit (RamII)) linked to a Bruker Vertex70 spectrometer. A continuous-wave Nd-YAG laser working at λ = 1064 nm was employed for excitation. A germanium detector operating at liquid nitrogen temperature was used. Raman scattering radiation was collected in a back-scattering configuration with a standard spectral resolution of 4 cm^−1^. The power of the laser beam was kept at a level lower than 50 mW in all cases. Around 2000–3000 scans were averaged for each spectrum to optimize the signal-to-noise ratio.

**Cyclic voltamograms** of **1**–**3** triindoles (c = 1 × 10^−3^ M) were recorded at a scan rate of 100 mV.s^−1^ in CH_2_Cl_2_/0.1 M Bu_4_NPF_6_ as electrolyte, using a Pt working electrode, a Ag/AgCl reference electrode and a Pt wire auxiliary electrode.

**Crystal Structure** determination of **2**. The crystals were selected under a polarizing optical microscope and glued on a glass fiber for a single-crystal X-ray diffraction experiment. Single-crystal X-ray data were obtained in a Bruker four circle kappa-diffractometer equipped with a Cu INCOATED microsource, operated at 30 W power (45 kV, 0.60 mA) to generate Cu Kα radiation (λ = 1.54178 Å), and in a Bruker VANTEC 500 area detector (microgap technology). Single crystal X-Ray diffraction data were collected by exploring over a hemisphere of the reciprocal space in a combination of φ and ω scans to reach a resolution of 0.86 Å, using a Bruker APEX3 [33] software suite (each exposure of 40 s covered 1º in ω). Unit cell dimensions were determined for least-squares fit of reflections with I > 20 σ. A semi-empirical absorption and scale correction based on equivalent reflection was carried out The structures were solved by direct methods. The final cycles of refinement were carried out by full-matrix least-squares analyses with anisotropic thermal parameters of all non-hydrogen atoms. The hydrogen atoms were fixed at their calculated positions using distances and angle constraints. All calculations were performed using APEX2 [33] software for data collection and data reduction and OLEX3 [34] to resolve and refine the structure, respectively. Appendix A summarizes the main crystal and refinement data for **2**. CCDC 2131157 contains the supplementary crystallographic data for **2**.

**Field effect transistors** were fabricated in a standard bottom gate-top contact architecture. First, the gate/dielectric substrates (Si/300 nm SiO_2_) were double-cleaned in an ultrasonic bath with ethanol prior to drying under a flow of nitrogen. Next, the surface was functionalized with a self-assembled monolayer of either octadecyltrichlorosilane (OTS) or hexamethyldisilazane (HMDS) to minimize interfacial trapping sites. Then, thin films of **1**–**3** were prepared by slow sublimation under vacuum conditions on preheated substrates at different temperatures, followed by an annealing treatment. Finally, 40 nm gold source and drain electrodes were thermally evaporated through a shadow mask. The devices were tested at room conditions by using an EB-4 Everbeing probe station with a 4200-SCS/C Keithley semiconductor characterization system.

**GIXDR** data using CuKα1 radiation was recorded by using a Bruker D8 DISCOVER diffractometer. The grazing incidence X-ray diffraction setup is equipped with a parabolic Göbel mirror and a conventional line focus Cu radiation tube (40 kV/40 mA). The Cu-kβ line is suppressed by a knife at the exit of the Göbel mirror. The nearly parallel incident beam was collimated with a 2 mm primary slit. No primary monochromator is used to improve the monochromaticity of the X-ray beam and to reduce the beam divergence after the Göbel mirror. Soller slits were used to reduce the axial divergence (divergence in the plane perpendicular to the diffraction plane) of the X-ray beam. This axial divergence is then reduced to 2.5°. The sample is mounted on a motorized goniometric head. Omega and chi scans are performed to align the sample along the incident beam. The z scan ensures that the sample is aligned along the incident beam with an error lower than 10 μm. The accuracy of the alignment of the sample obtained after performing different rocking curves is lower than 5/100°. To avoid any precession effect, the sample does not spin (no phi rotation). Since the sample is composed of nanometric grains, many grains are always in diffraction conditions even if the sample does not rotate. The diffracted beam is collected by a Scintillation detector. No radial Soller slits nor analyzer crystal is mounted in front of the detector. Measurements were made from 2 to 35° (2θ) during 120 min. The tube worked at 40 kV and 40 mA. The omega value was determined by several omega scanning at the two theta main peak of the triindole-based thin film.

**Atomic Force Microscopy (AFM):** thin films were recorded by a Multimode atomic force microscope with a Nanoscope V Controller (Bruker Corporation, Billerica, MA, USA) working in tapping mode.

**Computational method**: The optimum structures of the neutral and radical cation states of all the triindoles under study were performed at the framework of the Density Functional Theory (DFT) using the hybrid, generalized gradient approximation (GGA) functional B3LYP [35,36] and the hybrid meta-GGA functional of Truhlar and Zhao M06-2X [37] together with the 6-31G** [38,39] basis set, as implemented in the GAUSSIAN09 program [40]. All geometrical parameters were allowed to vary independently apart from the planarity of the rings, and no imaginary frequencies were observed, which ensures the finding of the global minimum energy. Calculations on radical cations were spin-unrestricted. Furthermore, vertical electronic excitation energies were performed by using the time-dependent DFT (TD-DFT) [41,42,43] approach on the resulting optimized geometries at the same level of theory. Absorption spectra were simulated through convolution of the vertical transition energies and oscillator strengths with Gaussian functions (0.3 eV width at the half-height) by using GaussSum 3.0 software [44]. Interestingly, both B3LYP and M06-2X functionals display similar trends in the molecular structure description (Appendix A), evolution of the energy gaps (Figure 3 in the main text when compared to Appendix A) and vertical electronic excitation characterization (Appendix A) when comparing the different compounds. Raman intensities were calculated by using an adjustment of the theoretical force fields in which the frequencies are uniformly scaled down by a factor of 0.968 to disentangle experimental misassignments. The theoretical spectra were obtained by convolution of the scaled frequencies and the Raman scattering activities with Gaussian functions (10 cm^−1^ width at the half-height). The reorganization energies were calculated directly from the relevant points on the potential energy surfaces by using previously reported standard procedures [45]. Transfer integrals for pairs of molecules taken out of the crystal structures of **1** and **2** and for cofacial dimers of **1**-**3** were calculated at the B3LYP/6-31G** level by using the approach described by Valeev et al. [46]. In this sense, HOMO and HOMO-1 were observed to be degenerated in energy on isolated molecules at the geometry of the crystal and, thus, the t_h_ values were calculated as [(t_HOMO.HOMO_^2^ + t_HOMO-1.HOMO-1_^2^ + t_HOMO-1.HOMO_^2^ + t_HOMO.HOMO-1_^2^)/4]^1/2^. The molecular orbitals’ distribution and vibrational eigenvectors were plotted using the ChemCraft 1.8 molecular modelling software [47]. For the geometry optimization of the trimer models, the B3LYP-D3 and M06-2X-D3 functionals, which include an explicit correction for dispersion [48], were used together with the ωB97XD [49] functional.

## 4. Conclusions

Two new crystalline *N*-trimethyltriindoles endowed with different substituents at 3,8 and 13 positions (either three methoxy or three acetyl groups) have been successfully synthesized and characterized. In addition, a comparison with the non-peripherally substituted derivative **1** in terms of how functionalization affects the resulting electronic and charge-transport properties and, ultimately, the OFET devices’ performance has been undertaken. Functionalization slightly affects both the HOMO electronic levels and the crystal packing, however it impacts the hole mobilities. The subtle interplay between the intramolecular and intermolecular properties has been rationalized with the help of DFT calculations. The insertion of three methoxy groups gives very highly ordered columnar structures, as in the non-substituted platform and lower intracolumnar distances; however, this results in the hole mobilities of the OFETs being lower by one order of magnitude (0.022 cm^2^ V^−1^ s^−1^ in **1** vs. 0.0014 cm^2^ V^−1^ s^−1^ in **2**). This counterintuitive result can be ascribed to lower hole transfer integral values due to the less efficient wave function overlaps between molecules in **2** despite their more planar backbones. On the other hand, the insertion of electron-withdrawing acetyl groups gives very poorly crystalline organization, showing the lowest hole mobilities within the series. To summarize, this experimental and theoretical study provides a clear picture of how the peripheral functionalization can impact the charge–transport of triindole-derivatives of importance to design new semiconductors with improved performance.

## Figures and Tables

**Figure 1 molecules-27-01121-f001:**
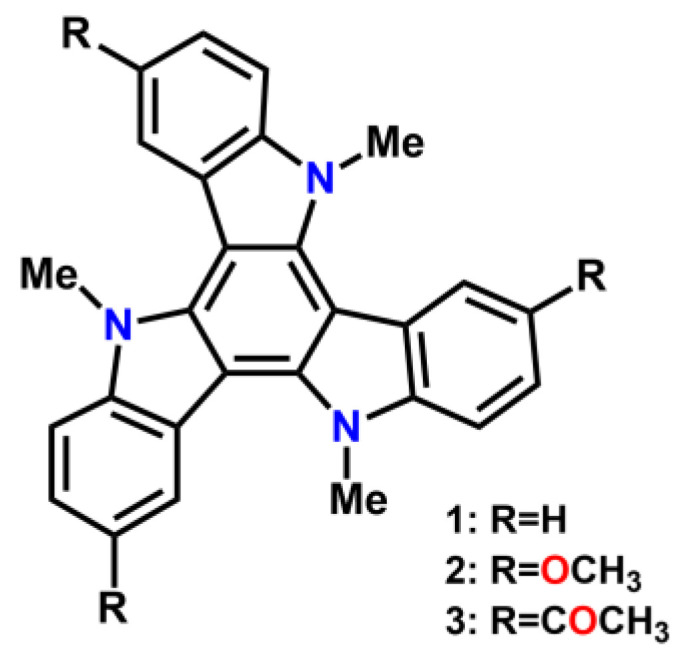
Chemical structures of the three triindole-based systems 1–3 reported in this work.

**Figure 2 molecules-27-01121-f002:**
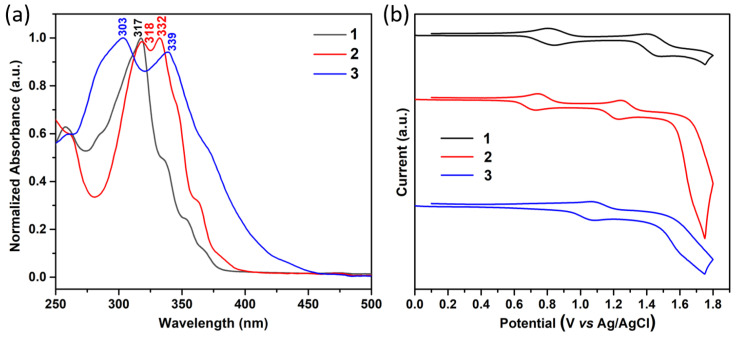
(**a**) Normalized absorption UV-vis spectra of **1**–**3** in CH_2_Cl_2_ at c = 5 × 10^−6^ molL^−1^; (**b**) cyclic voltammograms of **1**–**3** at c = 10^−5^ mol L^−1^ recorded in CH_2_Cl_2_/0.1 M TBAH using a Pt working electrode.

**Figure 3 molecules-27-01121-f003:**
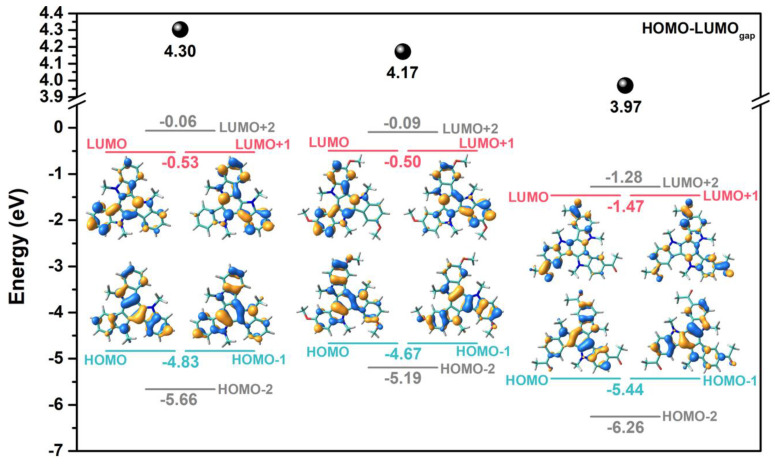
DFT-calculated molecular orbital energies and topologies for **1**–**3** compounds at the B3LYP/6-31G** level of theory.

**Figure 4 molecules-27-01121-f004:**
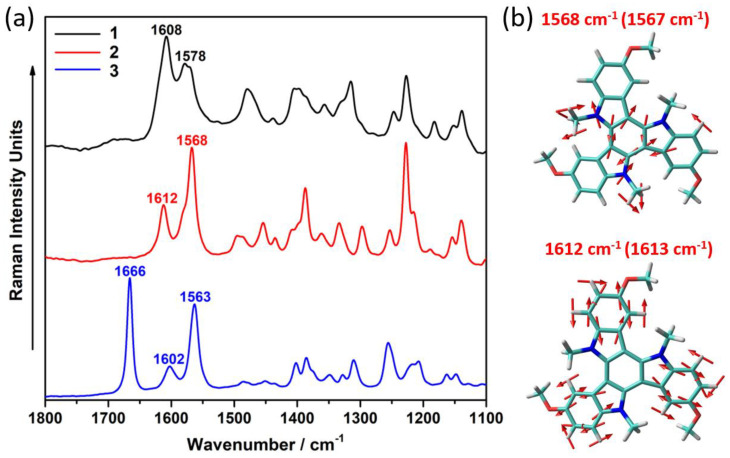
(**a**) Solid-state FT-Raman (λ_exc_ = 1064 nm) spectra for **1**–**3** compounds; (**b**) B3LYP/6-31G** vibrational eigenvectors associated with the most outstanding C=C/C−C Raman features of **2**, taken as representative example. The experimental and theoretical (in parentheses) wavenumbers are also shown.

**Figure 5 molecules-27-01121-f005:**
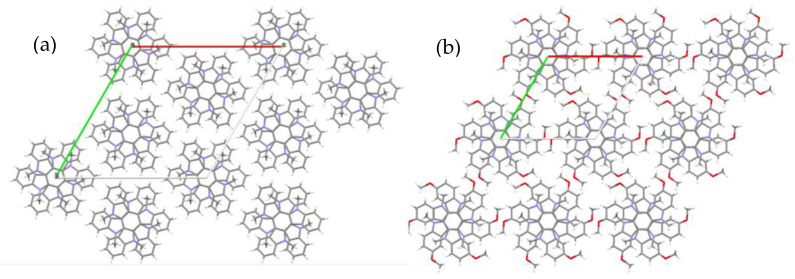
View of the packing of **1** (**a**) and **2** (**b**) along the crystallographic c axis.

**Figure 6 molecules-27-01121-f006:**
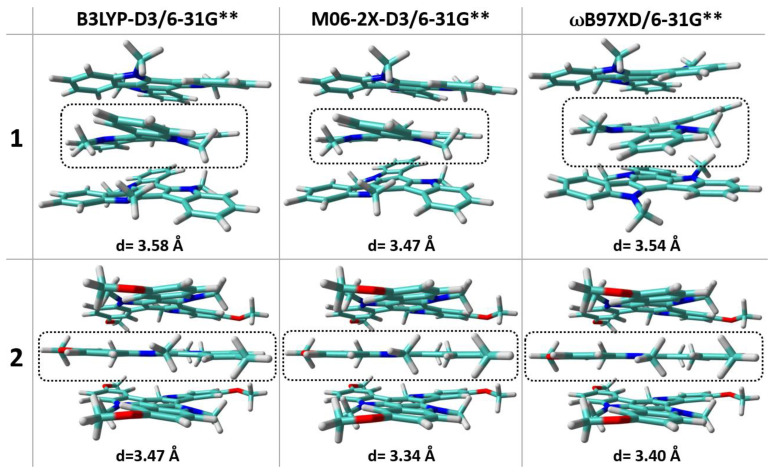
DFT-computed global minimum structures for a trimer model of **1** and **2**. The average centroid–centroid distance of the central aromatic rings between adjacent molecules is also shown.

**Figure 7 molecules-27-01121-f007:**
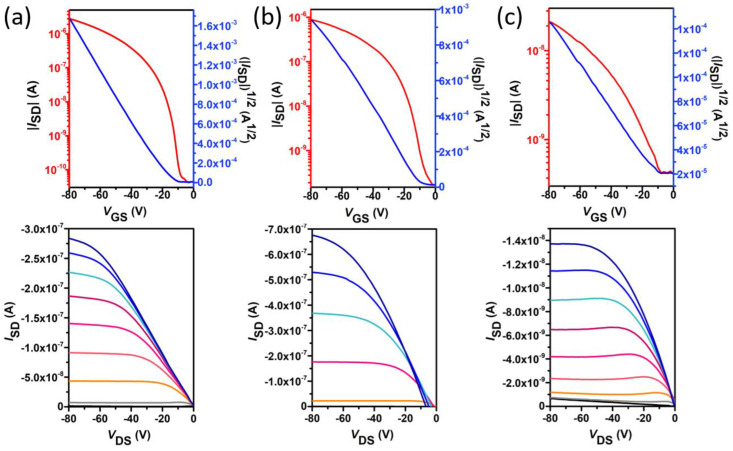
TFT transfer (**top**) and output (**bottom**) characteristics of triindoles **1** (**a**), **2** (**b**) and **3** (**c**). The transfer characteristics were measured at a constant source–drain voltage of −80 V. The output curves were measured at gate voltages varying from 20 to −80 V in intervals of 10 V.

**Table 1 molecules-27-01121-t001:** Main electrochemical properties of compounds **1**–**3**.

Compound	R	E_ox_ (V)	HOMO (eV) ^1^	LUMO (eV) ^1^
**1**	H	0.71	−5.08 (−4.83)	−1.72 (−0.53)
**2**	OCH_3_	0.69	−5.04 (−4.67)	−1.95 (−0.50)
**3**	COCH_3_	1.04	−5.39 (−5.44)	−2.73 (−1.47)

^1^ Values in parentheses refer to B3LYP/6-31G** theoretical data.

**Table 2 molecules-27-01121-t002:** OFET electrical data for vapor deposited film of **1**–**3** semiconductors measured at ambient conditions. Average and the best (in parenthesis) values are shown.

Compound	Deposition Conditions	*µ*_h_ (cm^2^ V^−1^ s^−1^)	V_T_(V)	I_ON_/I_OFF_
**1**	OTS, 60°	2.2 × 10^−2^ (2.8 × 10^−2^)	−13 (−17)	1 × 10^7^ (3 × 10^7^)
**2**	OTS, 90°	1.4 × 10^−3^ (1.6 × 10^−3^)	−4 (−8)	6 × 10^3^ (1 × 10^4^)
**3**	OTS, 120°	3.1 × 10^−5^ (4.5 × 10^−5^)	−17 (−21)	2 × 10^2^ (3 × 10^2^)

## Data Availability

Not applicable.

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
