# Peer review of "Functionalized Crystalline N-Trimethyltriindoles: Counterintuitive Influence of Peripheral Substituents on Their Semiconducting Properties"

_molecules, 2022, doi:10.3390/molecules27031121_

Round 1

Reviewer 1 Report

The authors report on the synthesis and characterization of three triindole derivatives with different functionalization, aiming to understand the origin of their charge transport properties. Even if the paper is potentially interesting for Molecules I believe that it should be improved in the following points:

1) The title cites a counterintuitive role of the functional groups on the charge transport properties. However by reading the paper it is not clear what is the not intuitive result found by the authors.

2)Considering that the indoles have been already investigated for several applications it shoud be useful to motivate the choice of the particular molecules that the authors realized.

3) The presentation of the absorption spectra is a bit confusing, as the authors provide the wavelength of the main peak, without clearly observing that the absorption edge is at much higher wavelength. In the following they report the absorption edge in energy, and it is thus not obvious that the absorption edge energy does not correspond to the absorption peak.

Moreover the authors claim that the experimental data are well reproduced by the DFT energy gaps. However it is evident, both from the figure S1 and from the data in Figure 3, that the DFT values are much higher than the absorption edge values and are qualitatively in the energy range of the hogh energy part of the spectra. A deeper comparison between theory and experiment is thus needed to shed light on this point, in particular evidencing the lack of consistence between the experimental and the theoretical values and/or trying to explain the origin of the absorption at energies below the main peak .

4) The discussion is difficult to follow, as the authors give names (actually yust numbers, 1, 2 and 3) to the molecules that are not easy to correlate to the structure along the paper. Moreover in the text they often refer to the molecules by citing the functional groups. I suggest to choose names that contains the functional group (or its behaviour ar acceptor or donor) and to use always the same names along the manuscript.

5) No data are shown on the crystallographic part, thus it is not clear that the compounds form crystals and how the authors determined which crystalline structure is present in the crystals.

6) The main part of the paper should be an understading of the origin of the charge transport properties. However the charge transport is typically strongly dependent on the film structure and packing. In this frame it is first of all not clear if the crystals discussed in the crystallography part are present in the films used in the transistors, or are instead spontaneously formed as an autput of the synthesis.

The second point is that the strong FET performances dependence on the annealing temperatures clearly evidence that the intermolecular properties are dominant. On one side this weakens the attempt to explain the data in terms of DFT calculations on the individual molecules. On the other side nothing is measured on the film properties after annealing (morphology and crystallinity), thus preventing to understand how the film properties affect the mobility.

Author Response

Reviewer 1

Comment: 1. The title cites a counterintuitive role of the functional groups on the charge transport properties. However by reading the paper it is not clear what is the not intuitive result found by the authors.

Answer: In this statement we refer to the fact that introduction of three methoxy groups gives rise to a similar columnar arrangement to that of parent N-trimethyltriindole but show shorter intrastacking distances (in agreement with more favorable CH-p interactions). However, the mobility values determined for this compound are lower, in spite that electronic coupling is highly dependence of the stacking distance. This unexpected behavior has been ascribed to less efficient wave function overlaps due to the HOMO wave function extension over the peripheral methoxy groups that result in lower transfer integrals between adjacent molecules, as determined with the help of density functional theory (DFT) calculations. We have rewritten the abstract and the introduction to clarify this point.  We hope the message is now clearer. 

Comment: 2. Considering that the indoles have been already investigated for several applications it shoud be useful to motivate the choice of the particular molecules that the authors realized.

Answer:  As mentioned in the introduction many triindole derivatives have been successfully incorporated in different electronic devices such as OFETs, solar cells, OLEDs. However, the derivatives reported in this manuscript are not intended to a particular application but to perform a systematic study on the influence of peripheral substituents in their solid-state organization, electronic propertiesand semiconductor behavior. Thus, please note that our main goal is to provide useful design principles to obtain new derivatives with improved semiconducting performance towards their incorporation in any of the above-mentioned devices.  

Comment: 3. The presentation of the absorption spectra is a bit confusing, as the authors provide the wavelength of the main peak, without clearly observing that the absorption edge is at much higher wavelength. In the following they report the absorption edge in energy, and it is thus not obvious that the absorption edge energy does not correspond to the absorption peak.

Moreover the authors claim that the experimental data are well reproduced by the DFT energy gaps. However it is evident, both from the figure S1 and from the data in Figure 3, that the DFT values are much higher than the absorption edge values and are qualitatively in the energy range of the high energy part of the spectra. A deeper comparison between theory and experiment is thus needed to shed light on this point, in particular evidencing the lack of consistence between the experimental and the theoretical values and/or trying to explain the origin of the absorption at energies below the main peak.

Answer: DFT calculations are used in this work to help to rationalize the optoelectronic and charge-transport properties of these triindole systems as a function of the different structural modifications. Therefore, a quantitative one-by-one assignment of the experimental/theoretical energy gap values is not the aim of this study. In any case, please note that experimental/theoretical energy gap value deviations are within the expected error obtained for DFT calculations using hybrid functionals, since isolated molecules without solvent effects have been considered in the simulations. Importantly, our DFT calculations nicely account for the evolution of the experimental HOMO-LUMO energy gap within the series.

In addition, TD-DFT calculations nicely reproduce the experimental absorption spectra, predicting an intense absorption band at lowest energy and several electronic transitions localized at higher energies below the main peak (experimentally observed around 250 nm) ascribed to electronic transitions between higher excited states with π-π* nature. The following discussion about the TD-DFT data have been included in the revised manuscript: 

“Time-dependent DFT (TD-DFT) vertical excitation energies reproduce quite nicely the experimental data, predicting a maximum absorption band localized at lowest energies and several electronic transitions around 250 nm with a π-π* nature (Figure S2)”.

Comment: 4. The discussion is difficult to follow, as the authors give names (actually yust numbers, 1, 2 and 3) to the molecules that are not easy to correlate to the structure along the paper. Moreover in the text they often refer to the molecules by citing the functional groups. I suggest to choose names that contains the functional group (or its behaviour ar acceptor or donor) and to use always the same names along the manuscript.

Answer:  Following the recommendation of the referee, we have replaced the formula of the functional groups by the names throughout the manuscript and unified the style of the nomenclature to facilitate the understanding of the discussion.

Comment: 5. No data are shown on the crystallographic part, thus it is not clear that the compounds form crystals and how the authors determined which crystalline structure is present in the crystals.

Answer:  Crystals of 2 were grown by slow evaporation of a CH2Cl2 solution as mentioned in the main text (page 6) and the details on the crystal structure determination could be found in the original version in the supporting information. Single crystals of 3 could not be grown, in spite of multiple attempts. In this revised version, we have moved the details on the crystal structure determination to the “materials and methods section” where the different characterization techniques used in this study are also detailed. A table including the main crystallographic and refinement data of 2 and powder X-ray diffractogram of 3 can be found in the supporting information.

Comment: 6. The main part of the paper should be an understading of the origin of the charge transport properties. However the charge transport is typically strongly dependent on the film structure and packing. In this frame it is first of all not clear if the crystals discussed in the crystallography part are present in the films used in the transistors, or are instead spontaneously formed as an autput of the synthesis.

The second point is that the strong FET performances dependence on the annealing temperatures clearly evidence that the intermolecular properties are dominant. On one side this weakens the attempt to explain the data in terms of DFT calculations on the individual molecules. On the other side nothing is measured on the film properties after annealing (morphology and crystallinity), thus preventing to understand how the film properties affect the mobility

Answer: We do agree with the referee that the intermolecular properties are crucial on the charge transport properties. Please, note that we have considered trimeric models to predict the intermolecular interactions (Figure 6) and dimeric structures to calculate the hole transfer integrals.

In terms of the film properties after annealing, we have added all the film characterization data in the SI (see morphologic characterization section in the SI). The Grazing Incidence X-Ray Diffraction (GIXRD) spectra of thin films of triindole 2 (Figure S12) and 3 (Figure S13) at different temperatures and the Atomic Force Microscopy (AFM) measurements of all compounds under study (Figures S14-S16) have been determined to explore the effect of the temperature on the resulting crystallinity of the film. Furthermore, we have included an additional paragraph in the revised manuscript aiming to better connect the resulting mobilities with the film properties:

“This fact is in total agreement with the progressive enhancement of crystallinity observed on the thermally treated devices based on triindole 2 (chosen as representative example) as determined by Grazing Incidence X-Ray Diffraction (GIXRD) and Atomic Force Microscopy (AFM) measurements (Figures S12 and S15). On the other hand, it was found that the performance of OTFT of 3 is relatively insensitive to thermal treatments (see Table S4), in consonance with the amorphous nature of the films under different annealing conditions (Figures S13 and S16)”.

Reviewer 2 Report

In this work, Sergio Gámez-Valenzuela et.al reported a series of N-trimethyltriindole derivatives functionalized with different substituents at 3, 8 and 13 positions. They investigated how the electronic nature of attached peripheral substituents influences on their electronic properties, crystal packing, and semiconductor behaviors. The interplay between the intramolecular and intermolecular properties has been rationalized with the help of DFT calculations. Although the materials’ semiconducting performance reported here is too low nowadays, through experimental and theoretical studies, a clear picture of the structure-property relationship has been established for this N-trimethyltriindole derivatives system. Overall, the work was carried out and the manuscript was well organized. I thus suggest accepting the manuscript after some revisions. Comments, which may help to further improve the work are provided below for the authors.

  1. 13C NMR of compound 3 is missing. For newly reported compounds, more detailed characterization like 13C NMR should be included, better also provide these spectra in SI for other researchers’ reference.
  2. For organic semiconductors, purity of materials is very important for realizing their true performance. As I mentioned, the hole mobilities for these compounds reported here is too low compared to current standards. Authors should exclude the purity effect by conducting detailed purity characterizations, that’s also related to the first issue need to be addressed.
  3. Regarding the TFT characterizations, transfer curve in Figure 7a seems to be not consistent with compound 1’s highest performance. Even threshold voltage is mismatched compared to Table 2. These data need to be further confirmed.
  4. Please double check everything before publishing, here listed a few typos found in this manuscript:

Possible wrong calculated values from Eox to HOMO in Table 1;

In line 245, “tt” should be “it”

In line 273, “establish”

Author Response

Reviewer 2

Comment: 1. 13C NMR of compound 3 is missing. For newly reported compounds, more detailed characterization like 13C NMR should be included, better also provide these spectra in SI for other researchers’ reference.

Answer: Compound 3 is very insoluble which makes it very difficult obtain a 13CNMR spectrum for this compound.  We have now accumulated the sample during 48h at 100ºC in C2D2Cl4 which allowed us to determine the corresponding peak values even though their intensity is still low. Copies of the 1HNMR and 13CNMR spectra of the two new compounds have now been included in the supporting information.

Comment: 2. For organic semiconductors, purity of materials is very important for realizing their true performance. As I mentioned, the hole mobilities for these compounds reported here is too low compared to current standards. Authors should exclude the purity effect by conducting detailed purity characterizations, that’s also related to the first issue need to be addressed.

Answer:  Please see answer to comment 1. On the other hand, we agree with the referee that the mobility values of this paper are moderate when compared with the current state of the art, however, we think that the great value of this paper is that it provides useful design principles to obtain new derivatives with improved semiconducting performance towards their incorporation in devices. 

Comment: 3. Regarding the TFT characterizations, transfer curve in Figure 7a seems to be not consistent with compound 1’s highest performance. Even threshold voltage is mismatched compared to Table 2. These data need to be further confirmed.

Answer: We have corrected figure 7a. A more representative transfer curve of data showed in Table 2 has been added in the revised manuscript.

Comment: 4.

Please double check everything before publishing, here listed a few typos found in this manuscript:

Possible wrong calculated values from Eox to HOMO in Table 1

Answer:

The HOMO levels of the three compounds have been estimated from the first oxidation potential referenced to the ferrocene/ferrocenium redox couple and considering a value of -4.8 eV for Fc with respect to zero vacuum level. We have now included this explanation in the manuscript. 

Comment: 5.

In line 245, “tt” should be “it”

In line 273, “establish”

Answer:

Following the reviewer’s suggestion, we have corrected these typos in the manuscript.

Round 2

Reviewer 1 Report

The authors revised the manuscript in order to address my previous comments. Overall the manuscript is improved and I appreciate the efforts of the authors.

There is anyway still an open point on the absorption spectra. First of all also in the revised manuscript part of the discussion is in peak wavelength and part in peak energy. This simply creates confusion.

Concerning the comparison with DFT it would be honest to add a comment on the discrepancy between the experimental absorption peak energy and the theoretical ones, eventually writing that the difference is within the typical values reported in literature. Otherwise it seems that the experiment and theory completely agree even if this is not true.

Author Response

As suggested by the reviewer, the discussion section of the absorption spectra has been modified in order to make clearer the distinction between the absorption edge and the absorption maximum. In addition, the optical gaps in the revised version are also reported in wavelenghts. On the other hand, regarding the difference between the theoretical and experimental data the following sentence has been included: " Note that redshift of the absorption maximum on going from 1 to 3 is well captured from the TD-DFT calculations being the difference within the typical values reported in literature [Chem. Soc. Rev. 2013, 42, 845]